# Verifying the Accuracy of 3D-Printed Objects Using an Image Processing System

Takuya Okamoto [1] and Sharifu Ura [2,*]

1    Graduate School of Engineering, Kitami Institute of Technology, 165 Koen-cho, Kitami 090-8507, Japan;
     d3227100016@std.kitami-it.ac.jp
2    Division of Mechanical and Electrical Engineering, Kitami Institute of Technology, 165 Koen-cho,
     Kitami 090-8507, Japan
*    Correspondence: ullah@mail.kitami-it.ac.jp

**Abstract:** Image processing systems can be used to measure the accuracy of 3D-printed objects. These systems must compare images of the CAD model of the object to be printed with its 3D-printed counterparts to identify any discrepancies. Consequently, the integrity of the accuracy measurement process is heavily dependent on the image processing settings chosen. This study focuses on this issue by developing a customized image processing system. The system generates binary images of a given CAD model and its 3D-printed counterparts and then compares them pixel by pixel to determine the accuracy. Users can experiment with various image processing settings, such as grayscale to binary image conversion threshold, noise reduction parameters, masking parameters, and pixel-fineness adjustment parameters, to see how they affect accuracy. The study concludes that the grayscale to binary image conversion threshold has the most significant impact on accuracy and that the optimal threshold varies depending on the color of the 3D-printed object. The system can also effectively eliminate noise (filament marks) during image processing, ensuring accurate measurements. Additionally, the system can measure the accuracy of highly complex porous structures where the pore size, depth, and distribution are random. The insights gained from this study can be used to develop intelligent systems for the metrology of additive manufacturing.

**Keywords:** 3D printing; computer-aided design; metrology; accuracy; image processing; error; porous structure

## 1. Introduction

Additive manufacturing, commonly known as 3D printing, fabricates objects by adding materials layer by layer [1–3]. (The authors use the phrases additive manufacturing and 3D printing interchangeably in this article.) The remarkable thing is that 3D printing holds great promise in transforming how we design and manufacture products [1,2]. Its unique capabilities provide an array of opportunities to explore innovative design ideas [1–3]. It has particularly emerged as a highly dependable manufacturing process for fabricating complex objects, including topologically optimized parts and porous structures [4–7]. Furthermore, this process has developed a reputation for effectively managing the production of small-volume, high-variety products. Nevertheless, inaccuracy may occur in 3D-printed objects due to pre-processing (e.g., file format conversion and slicing), material processing (e.g., shrinkage and temperature variation, filament blockage), post-processing (e.g., support removal and surface finishing), flaws in the materials (e.g., inconsistency in material properties and particle sizes), design issues (e.g., loose shells and thin walls), and machine structures (e.g., mechanical deformation). Section 2 briefly describes some relevant studies conducted to manage or eliminate the abovementioned causes.

The accuracy measurement ecosystem of 3D-printed objects consists of a set of system components, as schematically illustrated in Figure 1. The description of this ecosystem is

as follows: First, the CAD model of the object to be printed is sliced using an appropriate slicing system. This system provides the program to the control system of the 3D printer. The control system then runs the printer according to the program. Once the printer completes fabricating the object (printed object in Figure 1) or completes fabricating a given cross-section, an image acquisition system can be used to obtain the image of the printed cross-section. The obtained image then undergoes a series of image processing steps and produces a binary image denoted as a print image. Simultaneously, the target image generation system operates on the CAD model and produces a binary image of the desired design cross-section denoted as the target image. Finally, the accuracy elicitation system compares the print image with the target image pixel by pixel and determines how much the printed object conforms to the CAD model (see the comparison image in Figure 1).

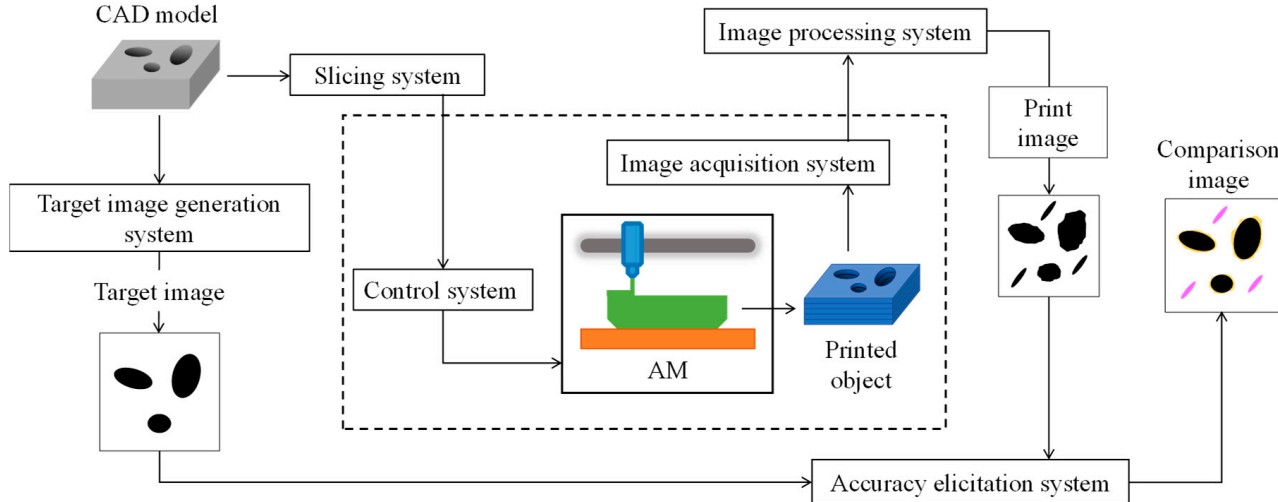

**Figure 1.** Systems for ensuring quality of 3D-printed objects.

The remarkable thing is that the image processing settings can significantly alter the print image and, thereby, the measurement results. Consequently, the measurement integrity relies on whether or not the image processing settings are selected correctly. This study focuses on this issue using a custom-made image processing system. (See Section 3 for the general description of the system.) The system first produces binary images of a given CAD model and its 3D-printed counterpart and then compares them pixel by pixel to quantify the accuracy. While processing the images, a user can select some vital parameters (e.g., the grayscale to binary image conversion threshold, noise reduction parameters, masking parameters, and pixel-fineness adjustment parameters). Depending on the settings of the parameters, the accuracy may vary a lot. Thus, the user must know which parameter settings alter the accuracy results and to what extent. Finally, the user can determine the optimal image processing settings, ensuring the integrity of accuracy measurement.

Based on the above contemplation, this article is written. For the sake of better understanding, the rest of this article is organized as follows: Section 2 presents a literature review on the accuracy-checking of 3D-printed objects. Section 3 presents the custom-made image processing system for investigating the accuracy measurement integrity of 3D-printed objects. Section 4 presents how to select the optimal image processing settings for relatively simple objects. Section 5 shows how to apply the system for determining the accuracy of a highly complex 3D-printed object (a porous structure with randomly sized and distributed pores). Section 6 presents the concluding remarks of this study.

## 2. Related Work

This section briefly describes some selected contributions regarding measuring the accuracy of 3D-printed objects.

Carew et al. [8] studied the accuracy of 3D modeling and 3D printing in forensic anthropology evidence reconstruction. They found that 3D printing accuracy is lower for complex parts, regardless of the printing process used, and printing layer resolution does not significantly affect accuracy, as modeling data resolution is usually higher. Other authors found similar results (e.g., Edwards et al. [9]). Lee et al. [10] used two different additive manufacturing processes to create a tooth replica. Depending on the additive manufacturing process, a replica can either shrink or enlarge, and its surface can become rough. They also found that the replica can be used in real-life applications despite accuracy problems because the accuracy remains within the stipulated tolerance limits. Leng et al. [11] developed a quality assurance framework to systematize the accuracy assessment of 3D-printed anatomic models. They found that three main areas cause inaccuracy: (1) image data acquisition, (2) segmentation and processing, and (3) 3D printing and cleaning. Both qualitative inspection and quantitative measurement are needed to assess the accuracy. The images of the 3D-printed model obtained by a high-resolution CT scanner can be compared with the original images to facilitate the quantitative measurement. George et al. [12] found that validated workflows improve 3D printing accuracy. However, software performance and manual adjustments can cause inaccuracies, impacting reproducibility. Any modification in workflows must be stepwise, with the help of STL dataset comparison metrics. New measurement methods are needed to achieve better results in evaluating the accuracy of 3D-printed medical models. Bortolotto et al. [13] employed a low-budget workflow consisting of 64-slice computed tomography (CT), three pieces of free and open-source software, and a commercially available 3D printer. They measured 3D-printed replicas and original objects using high-precision digital calipers and found that the dimensional inaccuracy is about 0.23 mm (0.055%), which is acceptable for medical applications. Herpel et al. [14] fabricated try-in dentures using milling (a subtractive manufacturing process) and 3D printing. The 3D printing was carried out at five facilities. Though the 3D-printed try-in dentures qualify for real-life application, they are less accurate than those produced by milling. Cai et al. [15] introduced the concept of residual STL volume as a metric to evaluate the accuracy and reproducibility of 3D-printed anatomic models. They applied the evaluation to maxillofacial bone and enhanced the accuracy of the 3D-printed structure. Kim et al. [16] studied the accuracy of a simplified 3D-printed implant for surgical guidance. They printed the same implant using three different additive manufacturing processes, namely photopolymer jetting (PolyJet), stereolithography apparatus (SLA), and multi-jet printing (MJP). They found that PolyJet and SLA can meet the required accuracy for clinical applications. Kwon et al. [17] studied the accuracy of a 3D-printed patient-specific implant. The shape datasets were extracted from CT images. The implants were fabricated using a 3D printer that uses photo-resin (curable under ultraviolet rays) with 0.032 mm resolution. In order to evaluate the accuracy, the implants were scanned using a micro-CT scanner, and the length and depth of the press-compressed and decompressed implants were compared using a Bland–Altman plot. The average differences in length were 0.67 mm $\pm$ 0.38 mm, 0.63 mm $\pm$ 0.28 mm, and 0.10 mm $\pm$ 0.10 mm. The average differences in depth were 0.64 mm $\pm$ 0.37 mm, 1.22 mm $\pm$ 0.56 mm, and 0.57 mm $\pm$ 0.23 mm, respectively. Yuan et al. [18] also obtained a similar degree of accuracy for the 3D-printed dental implants. Borgue et al. [19] considered that imperfections in material properties can lead to errors in 3D printing. They developed a fuzzy-logic-based approach for design for AM to manage uncertainties in material properties while meeting the quality standards of 3D-printed objects. Holzmond and Li [20] developed a system that detects two common 3D printing errors: filament blockages and low flow. They used a digital image correlation system to compare the point cloud captured from the printer-head movement program (g-code) and the point cloud of a printed surface in real time. Li et al. [21] considered that machine structure is the main cause of the error and developed an analytical model of the structure

of a given 3D printer to elucidate the printing error. Yu et al. [22] developed an image-processing-based approach to enhance the accuracy of 3D-printed microchannels. Using laser curing technology, they successfully modulated the optical proximity effect of curing light transmission and eliminated channel blockage and shape distortion while printing small-diameter channels. They used the local greyscale of the projection image as the 3D printing continued. Montgomery et al. [23] studied pixel-level grayscale manipulation to improve the accuracy of 3D printing based on digital light processing. They first printed an object according to the 2D binary image of the object. The grayscale image of the printed object was processed to create printing data (a relatively smooth contour). The processed information was used to print the same object with high accuracy. The method developed provided pixel-level grayscale control to create smooth features from sharp pixels. Ma et al. [24] developed an image-processing-based method for measuring the accuracy of layer-wise 3D food printing and identified the bottleneck (under- or over-extrusion). They first took a top-view image of the printed object (cookie), projected it on a vertical plane, and cropped it before it was segmented from its background using Ostu's automatic thresholding method [25]. They also converted the printer-head paths of each layer into a binary image. The image produced from printer-head paths and the image of the printed object were compared to quantify the accuracy. Vidakis et al. [26] developed a method that uses micro-computed tomography (micro-CT) images of 3D-printed objects to elucidate the dimensional and shape accuracies. They, however, did not show how the images were processed and compared. Eltes et al. [27] developed an image-processing-based accuracy-checking method for 3D-printed biomedical objects. They created surface meshes of the 3D-printed object using 3D scanning and compared them with the targeted surface meshes from CT scan images. The comparison was conducted using Hausdorff Distance (HD). Nguyen et al. [28] and others, e.g., see reference [29], developed a method to generate a model of a biomedical object processing sliced images from CT-scan data. The model was fabricated using 3D printers, and the CT-scan data of the printed object were obtained to check the accuracy. The details of the comparison mechanism that quantifies the accuracy were not presented. Xia et al. [30] developed an image acquisition and processing technique using a flatbed scanner to evaluate the shape accuracy of 3D-printed objects. The algorithms were formulated to extract useful shape information from the scanned images without human intervention. The centroid distance function and a root mean square error color map were used to visualize the inaccuracy effectively.

In synopsis, inaccuracy may occur in 3D-printed objects due to pre-processing (e.g., file format conversion and slicing), material processing (e.g., shrinkage and temperature variation, filament blockage), post-processing (e.g., support removal and surface finishing), flaws in the materials (e.g., inconsistency in material properties and particle sizes), design issues (e.g., loose shells and thin walls), and machine structures (e.g., mechanical deformation). Irrespective of the causes of inaccuracy, image processing is an effective means to measure the inaccuracy.

## 3. Proposed Image Processing System

As described in the previous section, the image of the target (or design) object and the image of the printed object are two valuable pieces of information by which one can guarantee whether or not the 3D-printed object is reliable. The real-time or offline comparison of these two types of images can provide valuable insights into the underlying manufacturing process and how to improve it. On the one hand, the target or design image can be created from the CAD model in STL format [3–5]. On the other hand, the image of the printed object can be created by applying some image processing techniques (e.g., converting a raw image to a grayscale image, converting a grayscale image to a binary image, removing noises from an image, masking and rescaling an image). To ensure the integrity of the accurate measurement results, it would be beneficial to develop a specialized image processing system that can examine how different image processing settings impact accuracy. This system could provide valuable insights into the extent of these effects and

help improve the accuracy of measurements in various applications. This section presents a custom-made image processing system that can fulfill the abovementioned measurement needs. In particular, this section presents the working principle, user interfaces, and performances of the proposed custom-made image processing system. The previous versions of the system can be found in references [31,32].

First, consider the working principle of the image processing system, as schematically illustrated in Figure 2. The left-hand side illustrations in Figure 2 show how the image processing system interacts with the data acquisition and CAD systems. The other side shows how the system processes images for the sake of comparison.

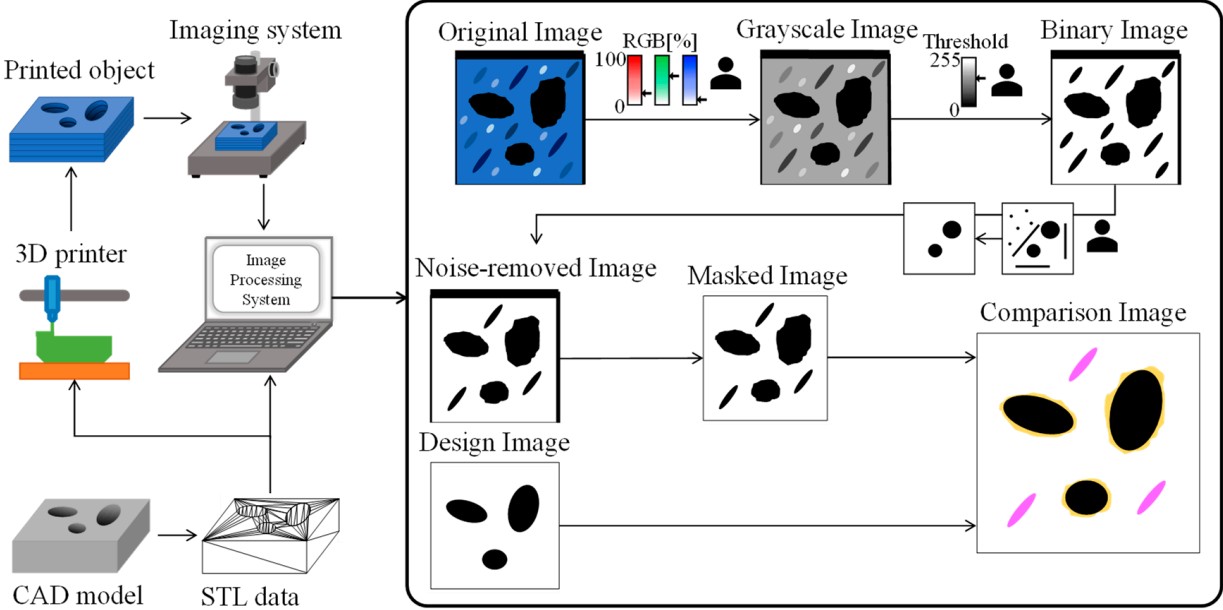

**Figure 2.** The working principle of the proposed image processing system.

As seen in Figure 2, a 3D printer operates using the STL data collected from a CAD model. After the printing operation, the printed object is collected. An imaging system (e.g., a microscope) can collect the image of the cross-section of the printed object. The image processing system acknowledges the raw image of the printed object as the original image. The system also acknowledges the STL data of the CAD model. The system generates the binary image of a given cross-section using the CAD model, denoted as the design image. The system processes the original image and produces a grayscale image using the user-defined values of RGB. The system then converts the grayscale image into a binary image using the user-defined threshold value. Afterward, the system removes the noise from the binary image using the user-defined values of the noise parameters and produces a noise-removed image. The system then applies a user-defined masking operation and produces the masked image. Finally, the design image is compared with the masked image, resulting in a comparison image. The comparison image is represented by pixels of four colors: black, white, violet, and yellow. Note that processes of obtaining a binary image, noise-removed image, and masked image can be reshuffled. For example, one can produce the masked image first before producing a noise-removed image. In that case, the noise-removed image is compared with the design image. For the sake of comparison, the resolution of the design image can be adjusted to the resolution of the masked, noise-removed, or binary image.

Let *B*, *W*, *V*, and *Y* denote black, white, violet, and yellow pixels, respectively. Let $P_D(i,j)$, $P_M(i,j)$, and $P_C(i,j)$, $i = 1, 2, \ldots, N$, $j = 1, 2, \ldots, M$, denote arbitrary pixels in the design image, masked/noise-removed/binary image, and comparison image, respectively. As such, $P_D(i,j) \in \{B, W\}$, $P_M(i,j) \in \{B, W\}$, and $P_C(i,j) \in \{B, W, V, Y\}$. The following rules are

maintained while generating a comparison image by comparing a design image with the masked, noise-removed, or binary image:

$$(P_D(i,j) = P_M(i,j) = B) \rightarrow P_C(i,j) = B \tag{1}$$

$$(P_D(i,j) = P_M(i,j) = W) \rightarrow P_C(i,j) = W \tag{2}$$

$$\Big((P_D(i,j) = B) \bigwedge (P_M(i,j) = W)\Big) \rightarrow P_C(i,j) = Y \tag{3}$$

$$\Big((P_D(i,j) = W) \bigwedge (P_M(i,j) = B)\Big) \rightarrow P_C(i,j) = V \tag{4}$$

Thus, violet or yellow pixels represent errors in a printing process, and minimizing the number of such pixels can lead to better quality outcomes. Consequently, printing error denoted as $E$ can be expressed as follows:

$$E = \left(\frac{YN + VN}{TN}\right)100\% \tag{5}$$

In Equation (5), $YN$ and $VN$ denote the numbers of yellow and violet pixels in the comparison image, respectively, and $TN$ denotes the total number of pixels in the same image.

*User Interfaces*

Based on the image processing system's outline described above, a system is developed. This sub-section presents the user interfaces of the developed systems. The system consists of five independent components denoted as follows: (1) grayscale–binary interface, (2) noise-removal interface, (3) masking–scaling interface, (4) target interface, and (5) comparison interface. The grayscale–binary interface, as shown in Figure 3, lets a user input an original image of a cross-section of a 3D-printed object. The user then sets the R, G, and B weights to convert the original image to a grayscale image. The user subsequently sets a value of the threshold (an integer in the interval (0, 255)) to obtain a binary image from the grayscale image. The interface displays the results. The user can save the grayscale and binary images in a preferred directory.

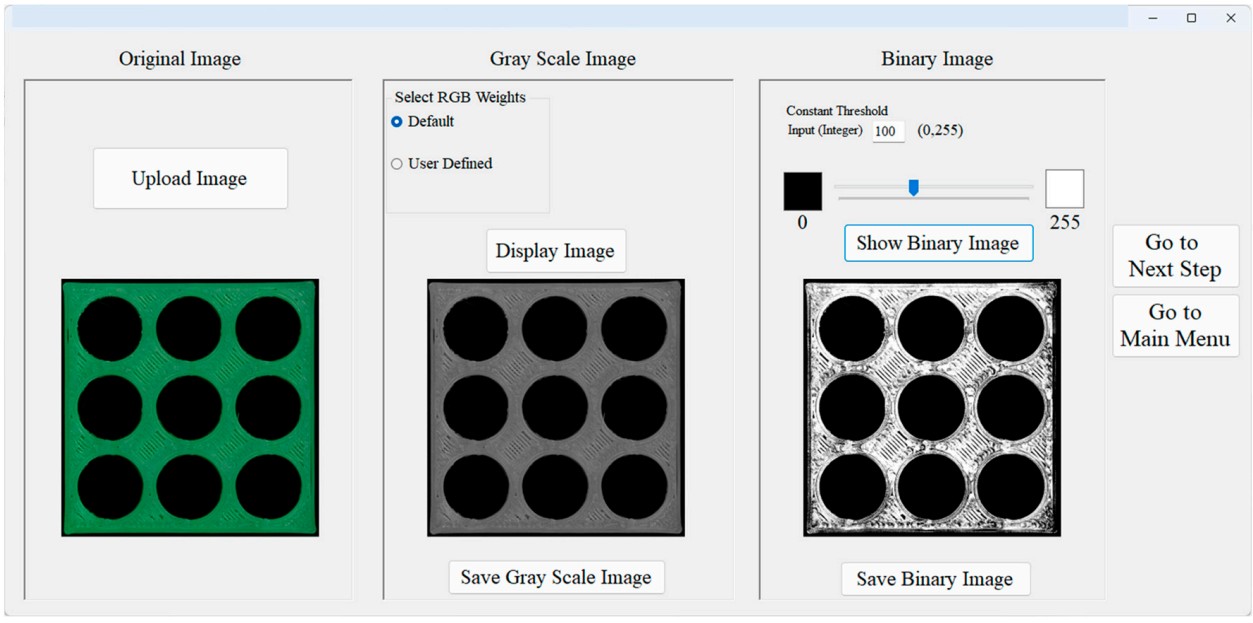

**Figure 3.** Grayscale–binary interface.

Using the noise-removal interface, as shown in Figure 4, a user can remove noises from a given binary image. The sources of noise are mostly the marks of the filaments and other irregularities on a given cross-section of the printed object. There are four parameters that help remove noises. The first two parameters are denoted as $\theta_{min}$ and $\theta_{max}$ (i.e., angles in degrees). They collectively set the orientation of the noises (i.e., the slopes of the noises) on the binary image. The other two parameters are critical spot size ($p_c$) and critical spot length ($w_c$). The units of these two parameters are pixels. The arbitrary case shown in Figure 4 corresponds to $\theta_{min}$ = 0° and $\theta_{max}$ = 50°, $p_c$ = 300 pixels, and $w_c$ = 100 pixels. As such, the system removes black spots whose size is less than or equal to 300 pixels, whose length is less than or equal to 100 pixels, and whose slope, in terms of an angle measured in degrees in the anticlockwise direction of the length, belongs to the angular range of [0°, 50°]. Note that the user can select black or white sports as noises. In the case shown in Figure 4, the user selects black spots as noises, which is the obvious thing to do. The interface displays the images before and after noise removal operations. The user can save the images in a preferred directory.

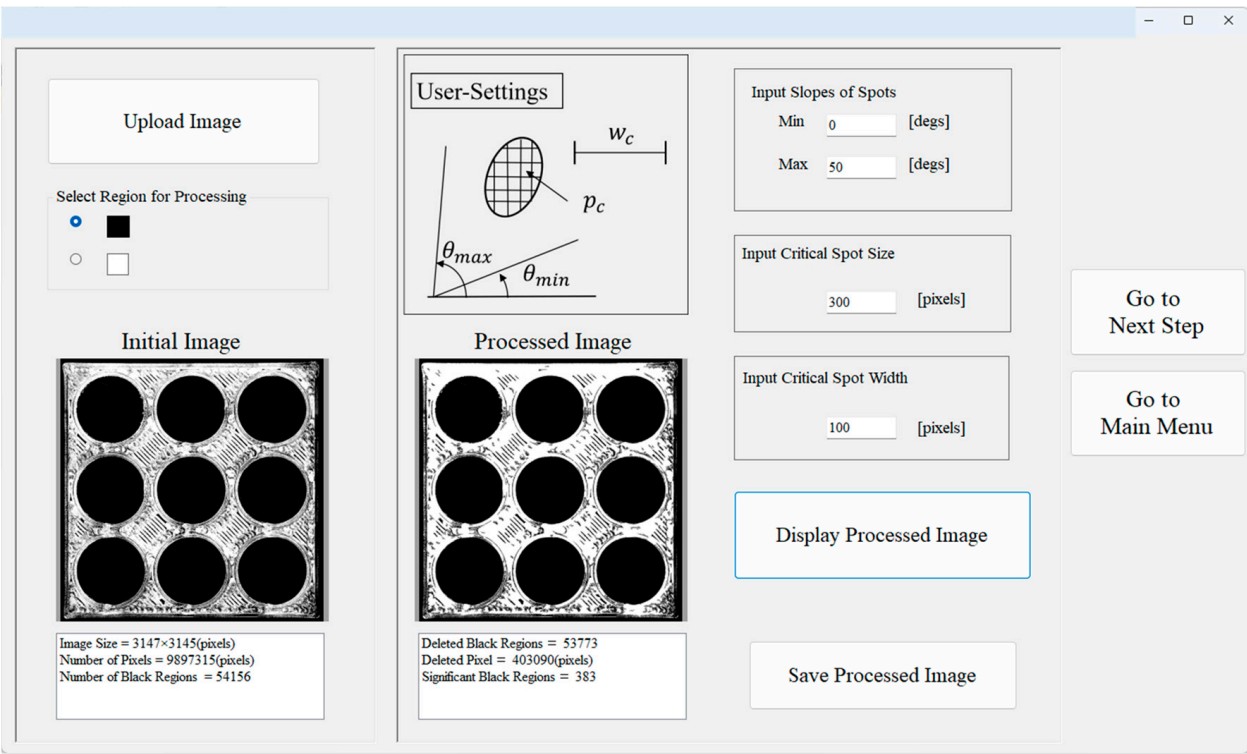

**Figure 4.** Noise-removal interface.

As shown in Figure 5, the masking–scaling interface is used to mask the unnecessary segment of a given binary image, preferably after removing the noises. In this interface, the size of the masked image is rescaled so that the pixel size of the design image (described below) matches that of the masked image. The case shown in Figure 5 presents a user cropping an image using a rectangular boundary so that it takes the size of the design image and the scale of the pixel matches that of the design image. The interface displays the images before and after masking and scaling operations. The user can save the images in a preferred directory.

As shown in Figure 6, the target interface creates a binary image of a given cross-section of the design. For this, the interface allows a user to input the STL data of the 3D CAD model (the design) of the object to be fabricated using a 3D printer. The user then inputs the height of the cross-section to produce the binary image. The interface displays the image. The user can save the image in a preferred directory.

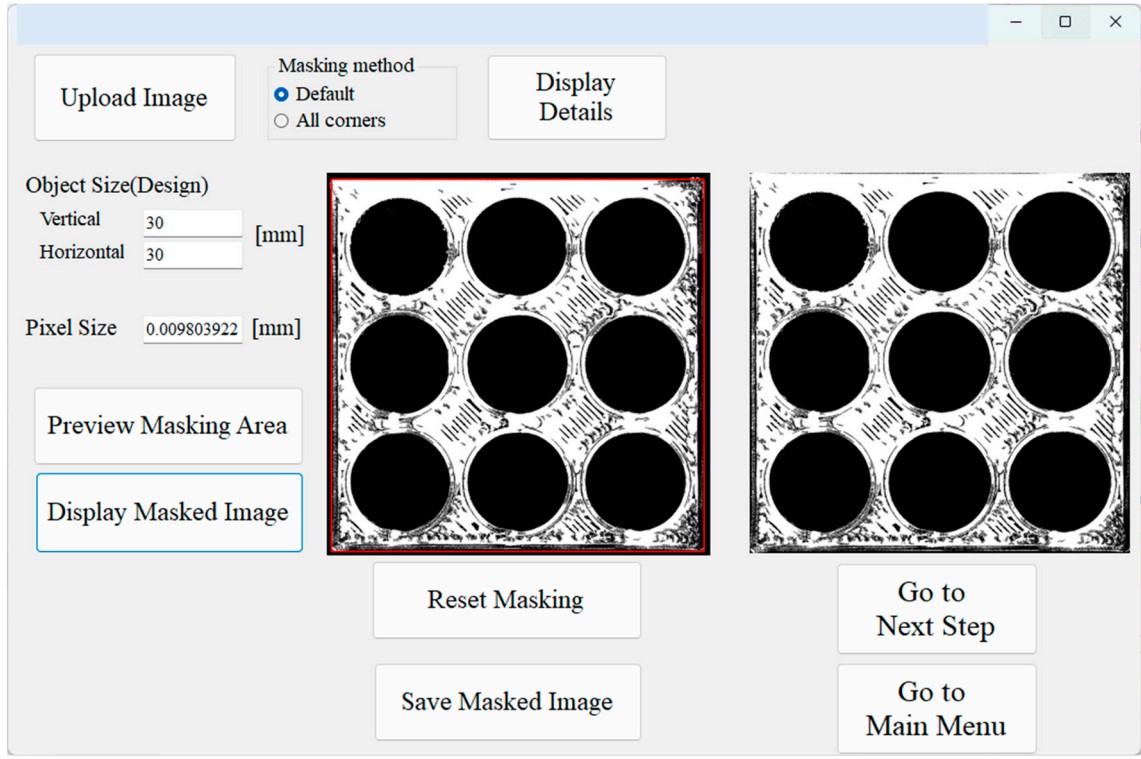

**Figure 5.** Masking–scaling interface.

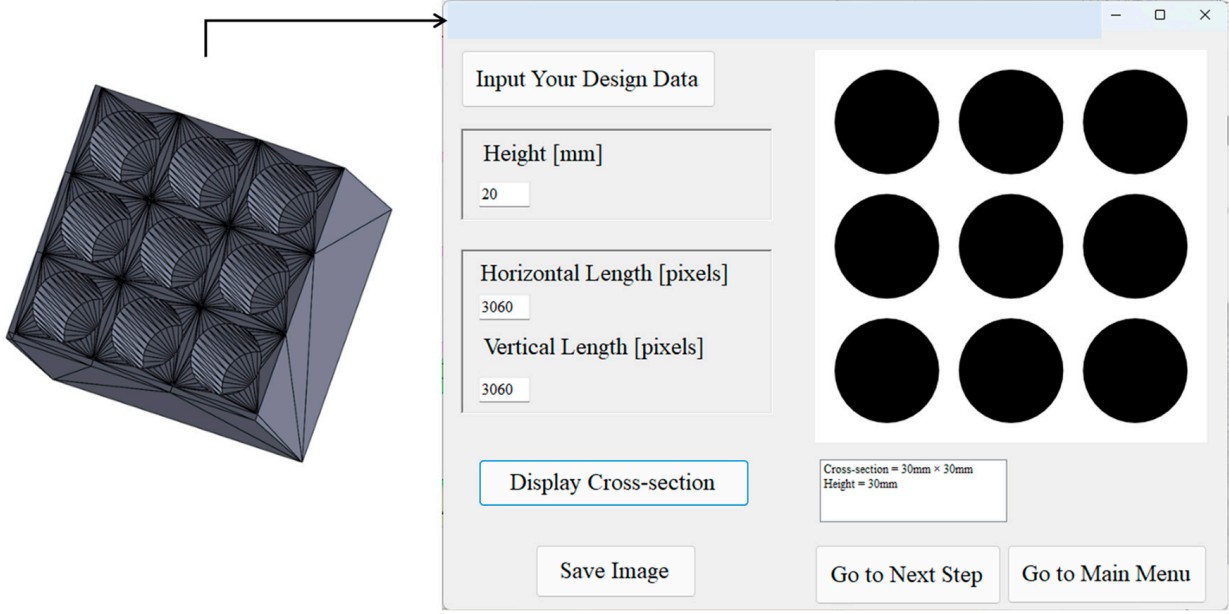

**Figure 6.** Target interface.

Finally, Figure 7 shows the comparison interface. In this interface, the user inputs the design image (i.e., the image created by the target interface, as shown in Figure 6) and the binary/noise-removed/masked image for comparison. The interface displays the comparison image with the four-color scheme described above. The interface helps the user save the comparison image in a preferred directory. The interface also displays error-related datasets ($YN$, $VN$, $TN$, and other relevant statistics).

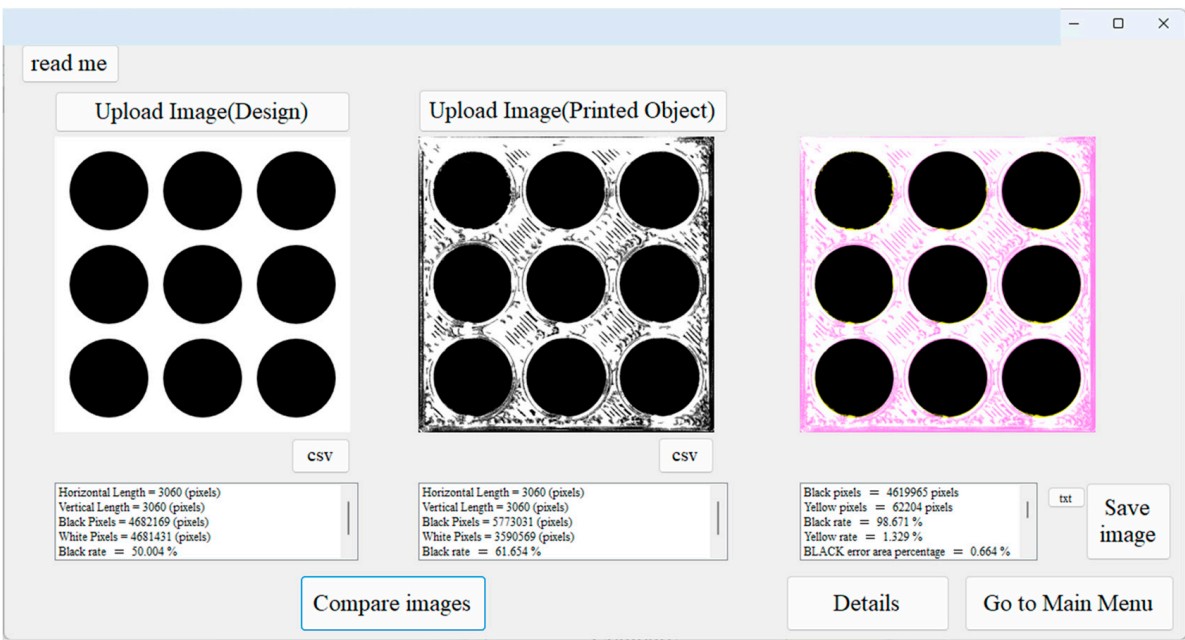

**Figure 7.** Comparison interface.

As mentioned earlier, the comparison image displays the errors in the 3D-printed object. Since it is produced by comparing it with a binary, noise-removed, or masked image, careful consideration is required when setting the parameters relevant to binary, noise-removed, and masked images. In particular, the settings of the threshold related to the binary image production process and the four parameters related to the noise removal process need careful consideration. See the arbitrary cases shown in Figures 8 and 9 for a better understanding. Figure 8 shows the original images of two objects printed in two different colors, green (a relatively dark color) and orange color (a relatively bright color). Both objects are printed using the same design (CAD model). The printing conditions for both objects are shown in Table 1.

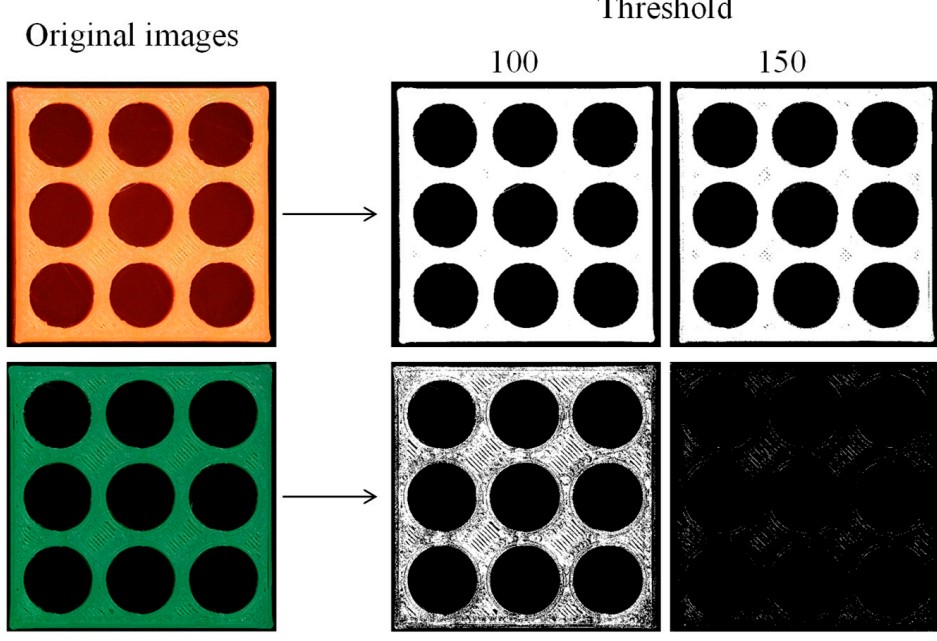

**Figure 8.** Effect of threshold.

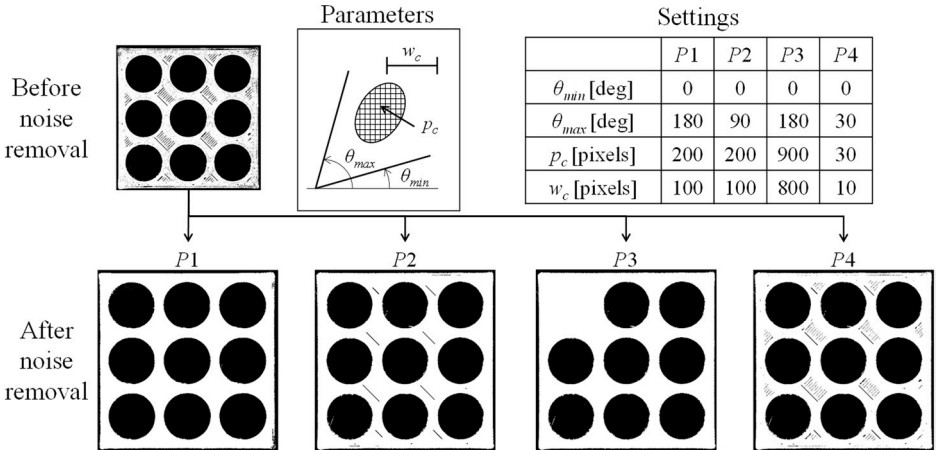

**Figure 9.** Effect of the settings of the parameters related to noise removal.

**Table 1.** Printing conditions.

| Items | Descriptions |
|---|---|
| Material | Thermoplastic Filament Made of Poly-Lactic Acid (PLA) |
| Printing technology | Fused Filament Fabrication (FFF) |
| Extrusion width [mm] | 0.4 |
| Extruder temperature [°C] | 205 |
| Printing speed [mm/s] | 50.0 |
| Infill speed [mm/s] | 80.0 |
| Layer height [mm] | 0.25 |
| Infill density [%] | 15 |
| Infill pattern | Grid |
| Infill angles [°] | 45, 135 |
| Printer | Raise3D Pro2™ (Irvine, CA, USA) |

From the greyscale images (not shown in Figure 8) of the original images, binary images are produced with the thresholds set at 100 and 150, respectively. For the orange object, both thresholds keep the information of the object. On the other hand, for the green image, both thresholds almost destroy the information of the object, with 150 being the worst. Consequently, a smaller threshold is preferable for dark colors, whereas a higher threshold is better for bright colors.

Figure 9 shows the effect of the settings of the parameters related to noise removal, i.e., two angles ($\theta_{min}$ and $\theta_{max}$) and two sizes ($p_c$ and $w_c$). As described before, the two angles collectively set the range of slopes within which the noises should be removed. Meanwhile, $p_c$ and $w_c$ set the critical spot (i.e., noise) size and length in terms of number of pixels. If the number of pixels of a spot is greater than $p_c$, this spot will not be removed during the noise removal process. The same argument is true for $w_c$. Figure 9 shows four different settings of $\theta_{min}$, $\theta_{max}$, $p_c$, and $w_c$, denoted as $P1$, $P2$, $P3$, and $P4$. The settings denoted as $P1$ remove a noise that consists of 200 pixels or less, has a length of 100 pixels or less, and can be oriented in any direction. Compared to $P1$, $P2$ puts a tight restriction on the orientation of a spot. Compared to $P1$, $P3$ puts less restriction on the sizes of a spot. Compared to $P1$, $P4$ puts a tight restriction on both the orientation and size. Comparing the images after applying the noise removal process, as shown in Figure 9, reveals that a less restrictive orientation of noise and a moderate noise size (corresponding to $P1$) can effectively remove noise, while other settings ($P2$, $P3$, and $P4$) may not yield the desired results.

## 4. Accuracy of a Simple 3D-Printed Object

This section presents a case study where the accuracy is estimated by comparing the design shown in Figure 6 with its 3D-printed counterparts shown in Figure 8. For better understanding, the first half of this section presents the results related to thresholds, and the other half presents results related to noise removal.

First, consider the results related to thresholds. In this case, binary images are produced for thresholds 10, 20, ..., 250 for both orange and green objects (Figure 8). Afterward, noises are removed using the same settings ($\theta_{min} = 0°$, $\theta_{max} = 180°$, $p_c = 200$, and $w_c = 100$). The noise-removed images are then processed as described in Section 3 to produce the masked images. The masked images are then compared with the design image (Figure 6) to produce the comparison images. Figure 10 shows four selected screen-prints of the comparison interface. As seen in Figure 10, the corresponding design image (see Figure 6) is compared with the masked images of the orange and green objects. The left-hand-side masked images correspond to threshold 40, whereas the right-hand-side images correspond to threshold 100.

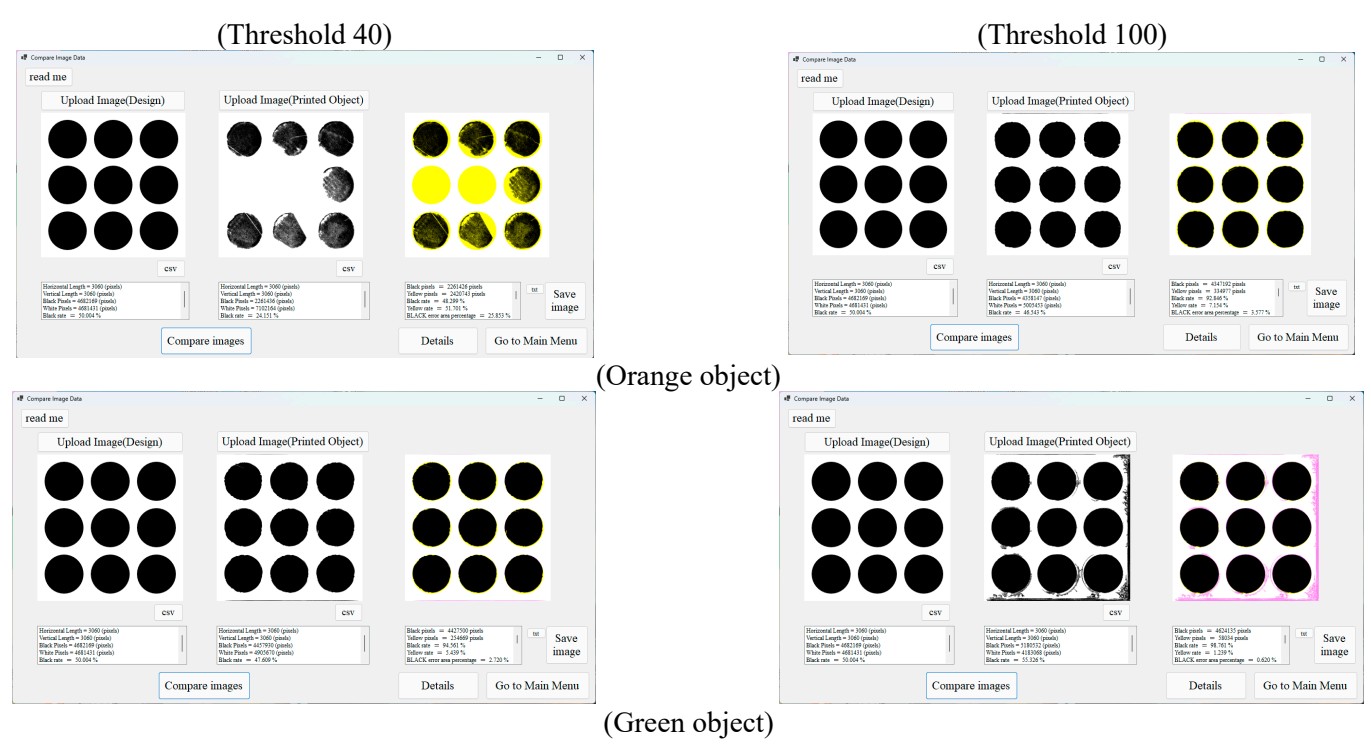

**Figure 10.** Screen-prints of the selected comparison interface.

However, Figure 11 shows the images of the orange object to be compared with the design image for thresholds 10, 20, ..., 250. As shown in Figure 11, for the thresholds 10 and 20, the images consist of white pixels only. With the increase in the threshold, the black pixels increase, and for thresholds 210 and above, the images consist of black pixels only. Figure 12 shows the comparison images corresponding to the images in Figure 11. As seen in Figure 12, yellow pixels decrease with the increase in the threshold, and violet pixels increase with the increase in the threshold.

On the other hand, Figure 13 shows the images of the green object to be compared with the design image for thresholds 10, 20, ..., 250. As shown in Figure 13, from threshold 70, black pixels start to dominate the images, and at threshold 130 and above, the images consist of black pixels only. Figure 14 shows the comparison images corresponding to the images in Figure 13. As seen in Figure 14, yellow pixels are rare, and the violet pixels increase with the increase in the threshold.

The errors are calculated by Equation (5) using the comparison images shown in Figures 12 and 14. The results are plotted in Figure 15. Figure 15a slows the overall trend in error, and Figure 15b shows the error trend for some selected thresholds. The green object's error is calculated for thresholds 70, 75, ..., 95, whereas the orange object's error is calculated for thresholds 140, 145, ..., 165. The comparison images for thresholds 75, 85, 95, 145, 155, and 165 are not shown in Figures 12 and 14, though others are shown. As seen in Figure 15, the orange object's error starts to decrease from threshold 30 and becomes minimal at around threshold 160. The minimal error is 2.161%. On the other hand, the error slowly decreases for the green object starting from threshold 10. This decreasing trend continues up to threshold 85. Afterward, the error increases sharply with the increase in the threshold. The minimal error here is slightly less than that of the orange object, which is 1.953%. Both objects exhibit the same error (around 50%) for very high thresholds. When the comparison image is either fully white or black, the expected error is about 50% because the design image consists of almost the same amount of black and white pixels. The plot in Figure 15a also exhibits the same result; for the white images (the first two images in Figure 12), the error is about 50%; for the fully black images (the last few images in both Figures 12 and 14), the error is about 50%. Thus, the image processing system presented in this study produces reliable results.

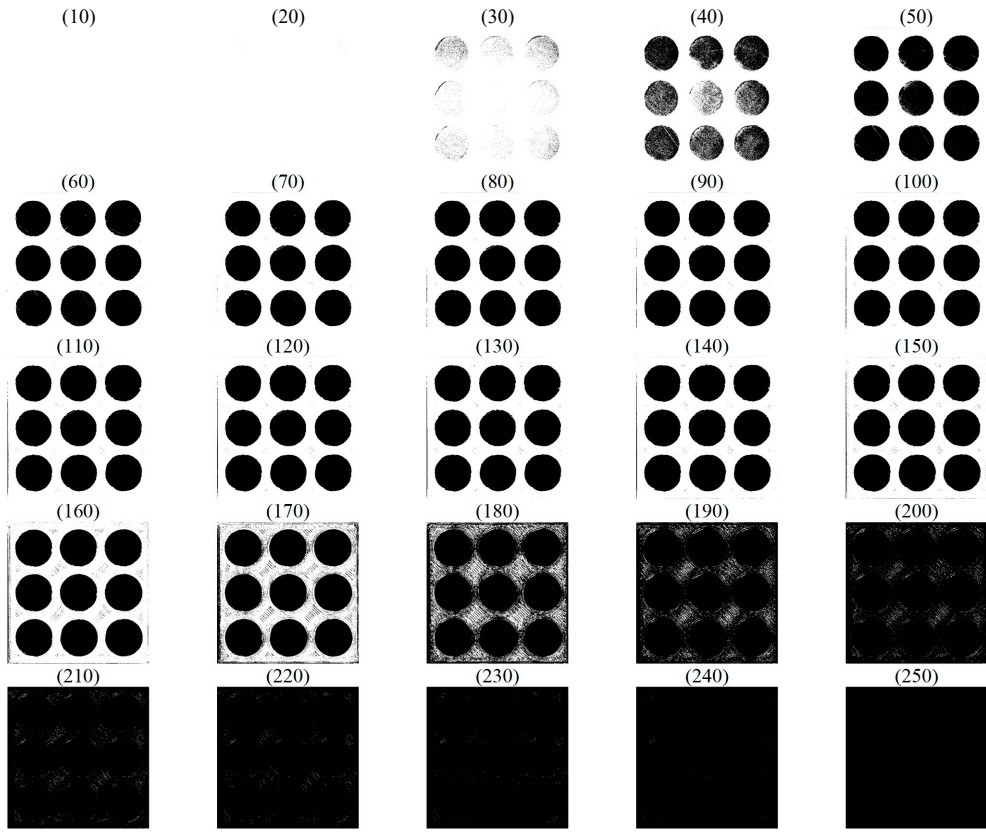

**Figure 11.** Images of the orange object for comparison. The integers in "()" are the thresholds.

What if the noise removal conditions are varied at the optimal threshold? In order to answer this question, nine sets of noise removal conditions denoted as 1, ..., 9 are considered for each object, as shown in Table 2. The conditions consider a constant threshold equal to the optimal threshold for each object (green or orange). The results are summarized in a plot, as shown in Figure 16. As seen in Figure 16, the error increases slightly when the noise orientation parameters are kept somewhat tight. When noise orientation parameters are kept wider, the error remains minimal and is not affected by parameters of noise size shown in see Table 2.

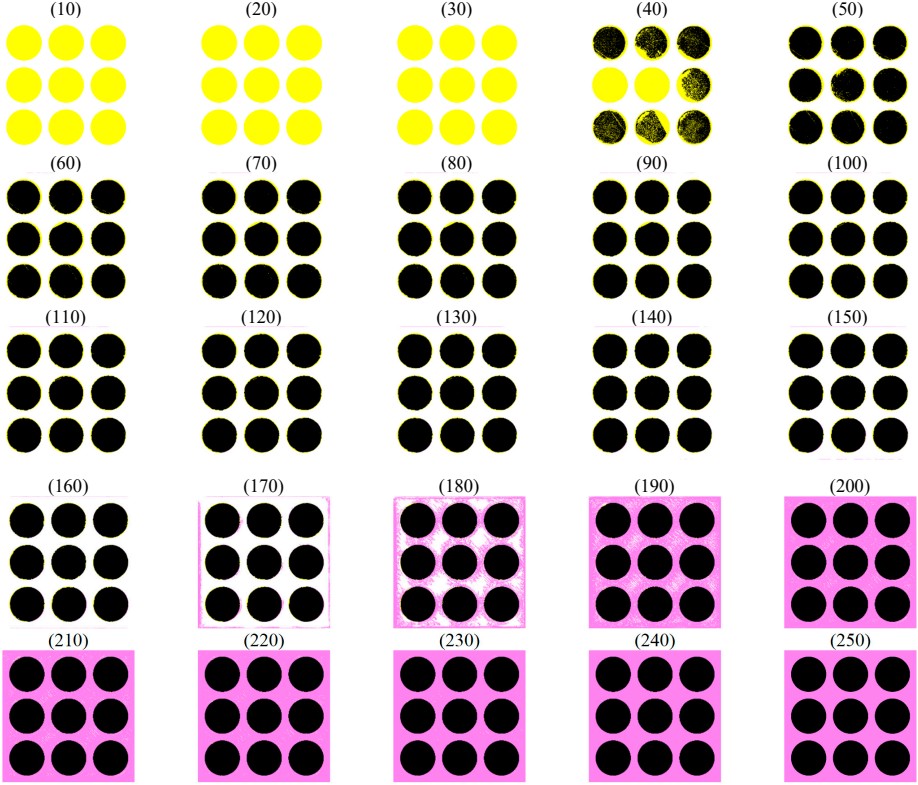

**Figure 12.** Comparison images of the orange object. The integers in "()" are the thresholds.

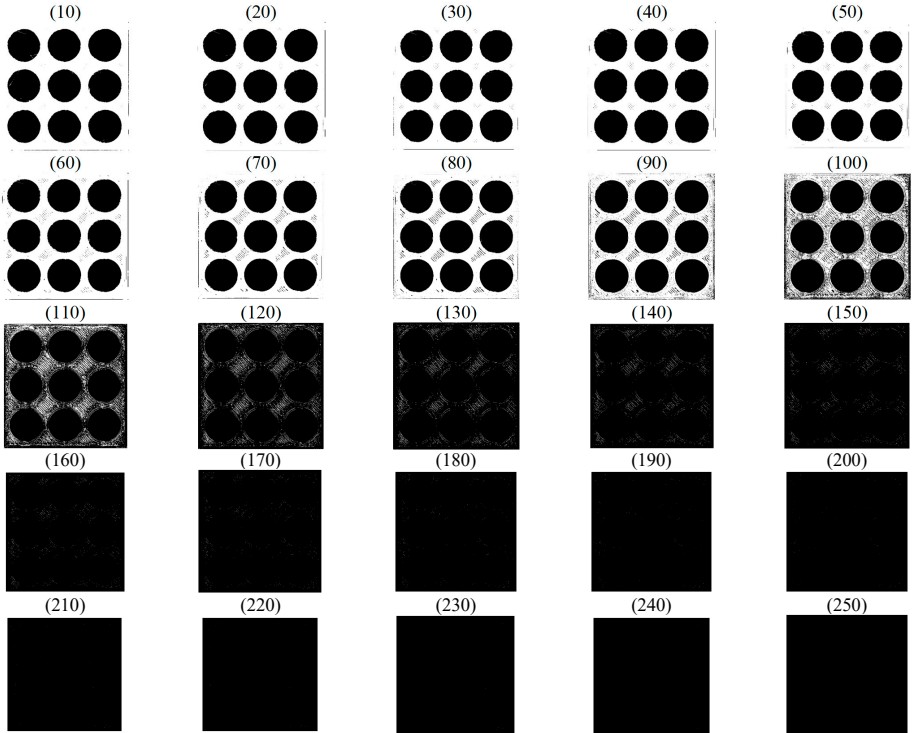

**Figure 13.** Images of the green object for comparison. The integers in "()" are the thresholds.

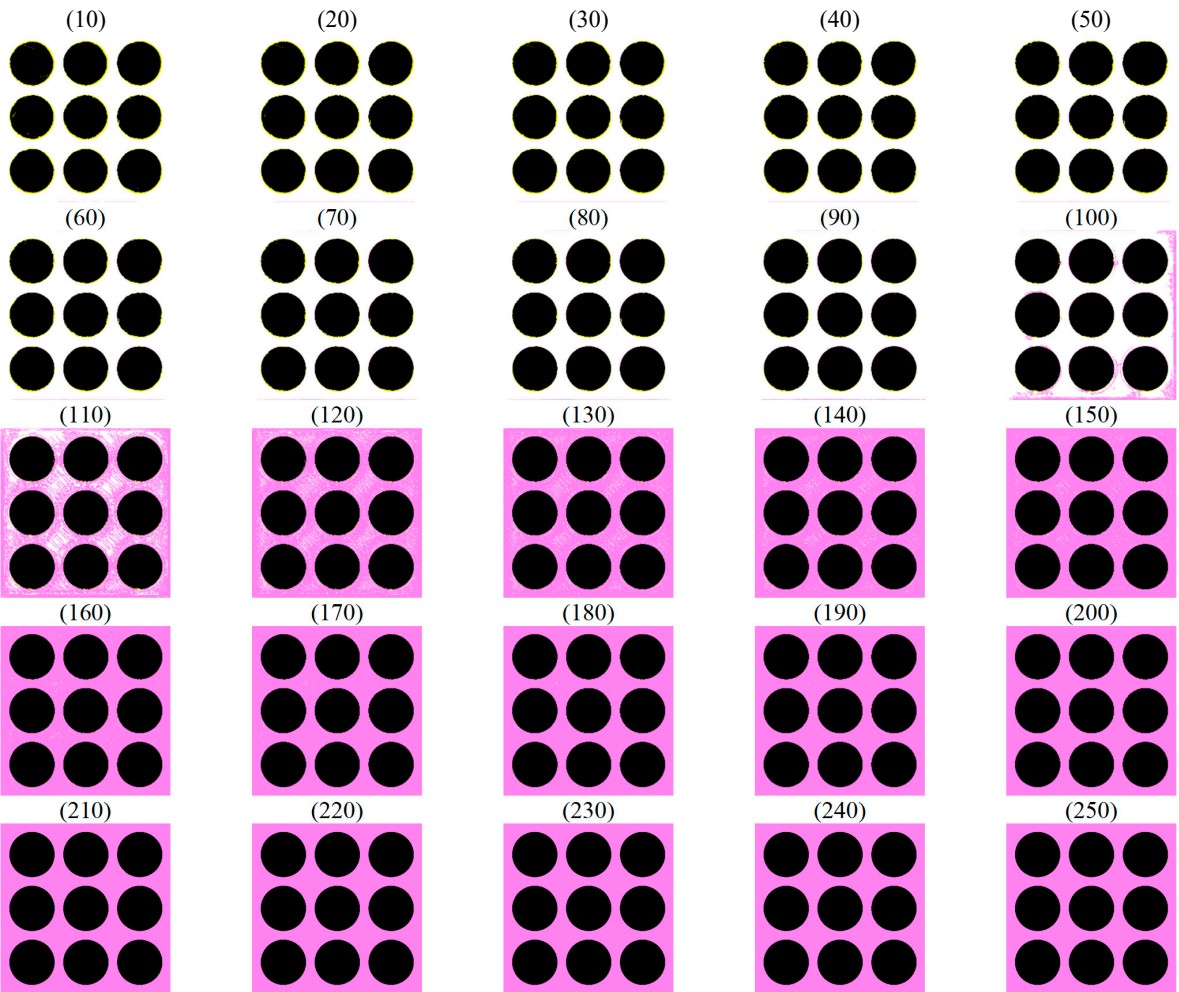

**Figure 14.** Comparison images of the green object. The integers in "()" are the thresholds.

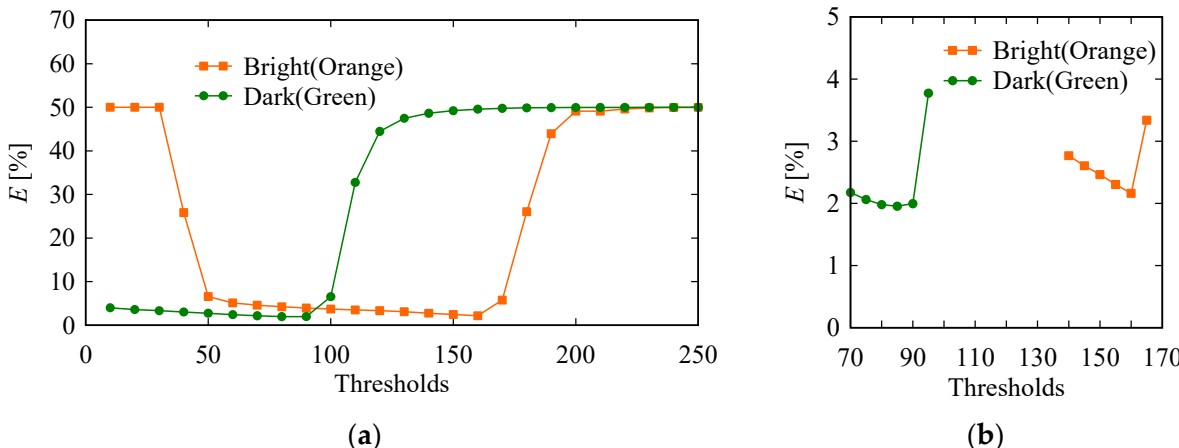

**Figure 15.** Error estimation for the orange and green objects. (**a**) Overall error trend. (**b**) Error trend for selected thresholds.

**Table 2.** Nine sets of conditions for the error analysis.

| Conditions | $\theta_{min}$ [°] | $\theta_{max}$ [°] | $p_c$ [Pixels] | $w_c$ [Pixels] | Thresholds |
|---|---|---|---|---|---|
| 1 | | | 100 | 50 | |
| 2 | | 45 | 200 | 100 | |
| 3 | | | 500 | 250 | For the orange object 160 |
| 4 | 0 | | 100 | 50 | For the green object 85 |
| 5 | | 90 | 200 | 100 | |
| 6 | | | 500 | 250 | |
| 7 | | | 100 | 50 | |
| 8 | | 180 | 200 | 100 | |
| 9 | | | 500 | 250 | |

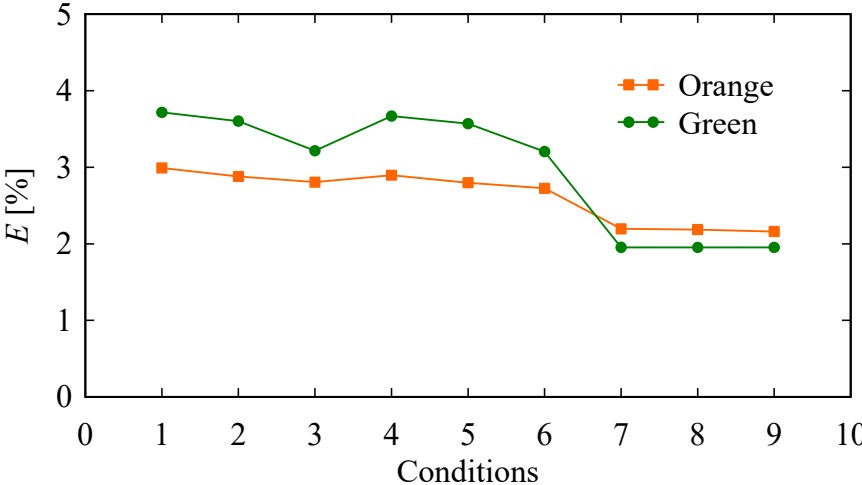

**Figure 16.** Variability in the error due to different noise removal settings.

In synopsis, if the underlying 3D printing operations remain normal, then approximately 3% error can be expected from the proposed image-processing-based accuracy measurement method, provided that the optimal threshold for the given color, wider noise orientation, and moderate noise sizes are selected. The threshold significantly impacts the error estimation process compared to other parameters.

## 5. Accuracy of a 3D-Printed Porous Structure (Complex Object)

As mentioned before, 3D printing has revolutionized the process of fabricating complex objects such as topologically optimized structures and porous structures, paving the way for innovative solutions in various fields and industries [4–7]. Unlike simple objects, complex objects may consist of loose shells, thin walls, and other difficult-to-fabricate features, resulting in sustainable inaccuracy [31,32]. Thus, measuring the accuracy of 3D-printed complex objects has become a critical issue. This section uses the presented image processing system to determine the accuracy of a complex object, i.e., a porous structure consisting of randomly sized and distributed pores.

Figure 17 shows the CAD model and 3D-printed counterpart of the porous specimen used in this study. In particular, Figure 17a shows a triangulation view of the CAD model of the porous specimen. Figure 17b shows an orthogonal view of a CAD model of the porous specimen. Figure 17c shows a picture of a 3D-printed counterpart of the porous specimen. As seen in Figure 17, the specimen is a rectangular prism with randomly sized and distributed pores. It is designed using the system shown in [5]. The outer dimensions

of the structure are 30 mm × 30 mm × 30 mm. It is printed using the specifications shown in Table 1, except for the infill-related parameters. This time, the infill rate is kept at 100%. As a result, the infill pattern and infill angle are not relevant here. The filament color is orange. The printing process is interrupted at heights 1 mm, 5 mm, 10 mm, 15 mm, 20 mm, and 25 mm to take images of the cross-section of the printed object at 1 mm, 5 mm, 10 mm, 15 mm, 20 mm, and 25 mm. The images are shown in Figure 18. As seen in Figure 18, the structure exhibits highly complex pores at each height, and complexity differs from one height to another. The images shown in Figure 18 are processed to measure the accuracy of the porous specimen. The results are summarized in Figure 19. Figure 19a shows the original images of the specimen at heights 1 mm, 5 mm, 10 mm, 15 mm, 20 mm, and 25 mm, which are already shown in Figure 18. These images are processed using the presented system. The optimal image processing settings for an orange-colored object described in the previous section are used to obtain the binary images for comparison. The resulting images of the printed specimen for comparison are shown in Figure 19b. The CAD model (Figure 17) of the specimen is processed at heights 1 mm, 5 mm, 10 mm, 15 mm, 20 mm, and 25 mm to produce the design images for comparison, as shown in Figure 19c. The respective comparison images are shown in Figure 19d. The errors exhibited by the comparison images are summarized in Table 3. This time, the minimal error is 3.85%, which corresponds to a height of 1 mm. The maximum error is 10.89%, which corresponds to a height of 10 mm. The average error is 8.695%, and the standard deviation is 2.803%. It means that the printing error increased three times due to the complexity of the design compared to that of the simple design. Note that for heights 10 mm, 15 mm, and 20 mm, the noises on the boundaries could not be removed properly. These remaining noises are the cause of the high values of noise for these heights. The error could have been much smaller if the noise had been removed properly.

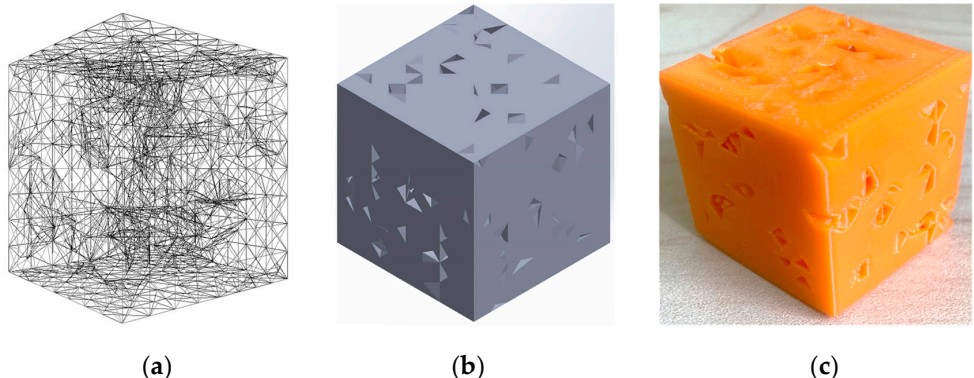

|       (a)       |       (b)       |       (c)       |

**Figure 17.** CAD model and 3D-printed counterpart of the porous specimen. (**a**) Triangulation view of the CAD model of the porous specimen. (**b**) Orthogonal view of the CAD model of the porous specimen. (**c**) 3D-printed counterpart of the porous specimen.

**Table 3.** Error in the selected layers of the porous structure.

| Heights [mm] | 1 | 5 | 10 | 15 | 20 | 25 |
|---|---|---|---|---|---|---|
| *E* [%] | 3.85 | 9.29 | 10.89 | 10.50 | 10.74 | 6.90 |

What if the images of the printed specimen are obtained using a micro-CT scanning device? Exploring this possibility is a valid option. Therefore, the following case study is conducted.

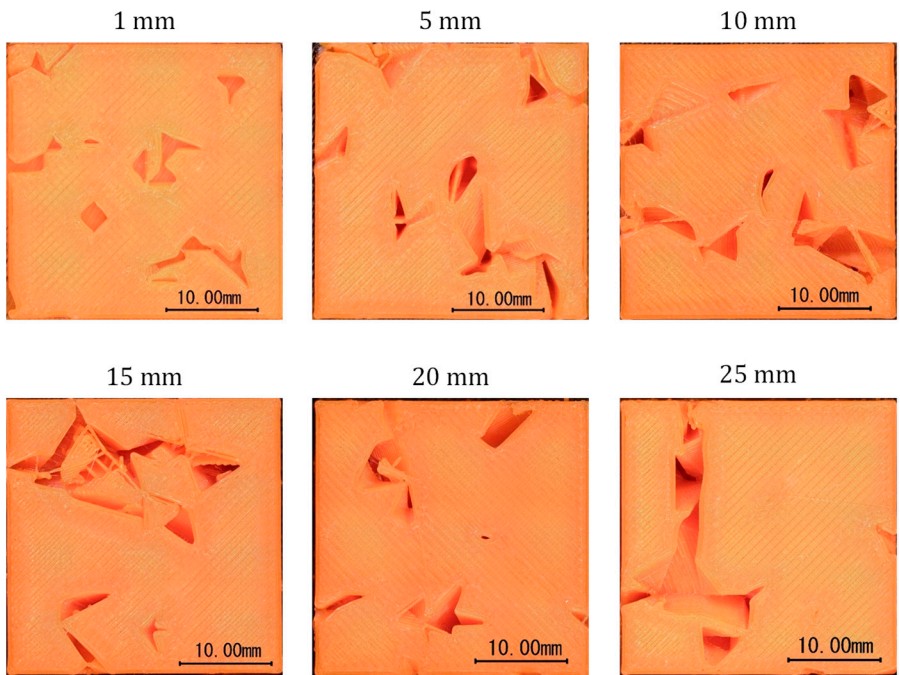

**Figure 18.** Original images of the porous specimen at different heights.

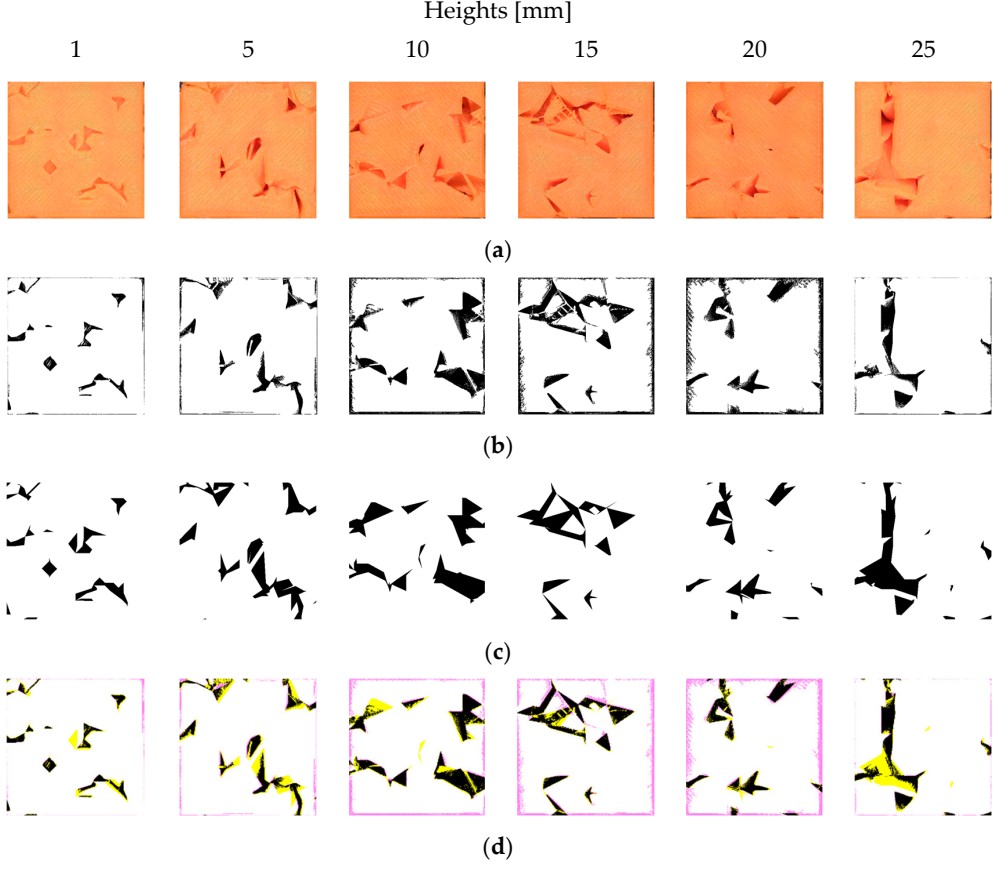

**Figure 19.** Different images of the porous structure relevant to accuracy assessment. (**a**) Original images of the specimen at different heights. (**b**) Processed images of the specimen for comparison. (**c**) Design images. (**d**) Comparison images.

Figure 20a shows images of the cross-sections on the planes *x-y*, *y-z*, and *z-x* of the CAD model of the specimen. The image of the cross-sections on the *x-y* plane corresponds to *z* = 15 mm. On the other hand, Figure 20b shows images of the cross-section on the planes *x-y*, *y-z*, and *z-x* of the printed specimen (Figure 17c). The cross-section image on the *x-y* plane corresponds to *z* = 15 mm. The images are obtained using a micro-CT scanning device. The presented image processing system can process micro-CT scans of specimen cross-sections. Accordingly, Figure 21 shows the results. Figure 21a shows the images of the specimen at heights 1 mm, 5 mm, 10 mm, 15 mm, 20 mm, and 25 mm. The images are obtained by scanning the specimen (Figure 17c) using a micro-CT scanning device. These images are processed using the presented system. Since color is not an issue here, the images are processed using the presented image processing devices where the settings are as follows: threshold = 100, $\theta_{min}$ = 0°, $\theta_{max}$ = 180°, $p_c$ = 40, and $w_c$ = 20. The resulting images of the printed specimen for comparison are shown in Figure 21b. The CAD model (Figure 17a,b) of the specimen is processed at heights 1 mm, 5 mm, 10 mm, 15 mm, 20 mm, and 25 mm to produce the design images, as shown in Figure 21c. The respective comparison images are shown in Figure 21d. The errors exhibited by the comparison images are summarized in Table 4. This time, the minimal, maximal, and average errors are 2.18%, 4.06%, and 3.22%, respectively, and the standard deviation is 0.655%. This time, the noise removal process works properly for all cross-section images, resulting in smaller errors.

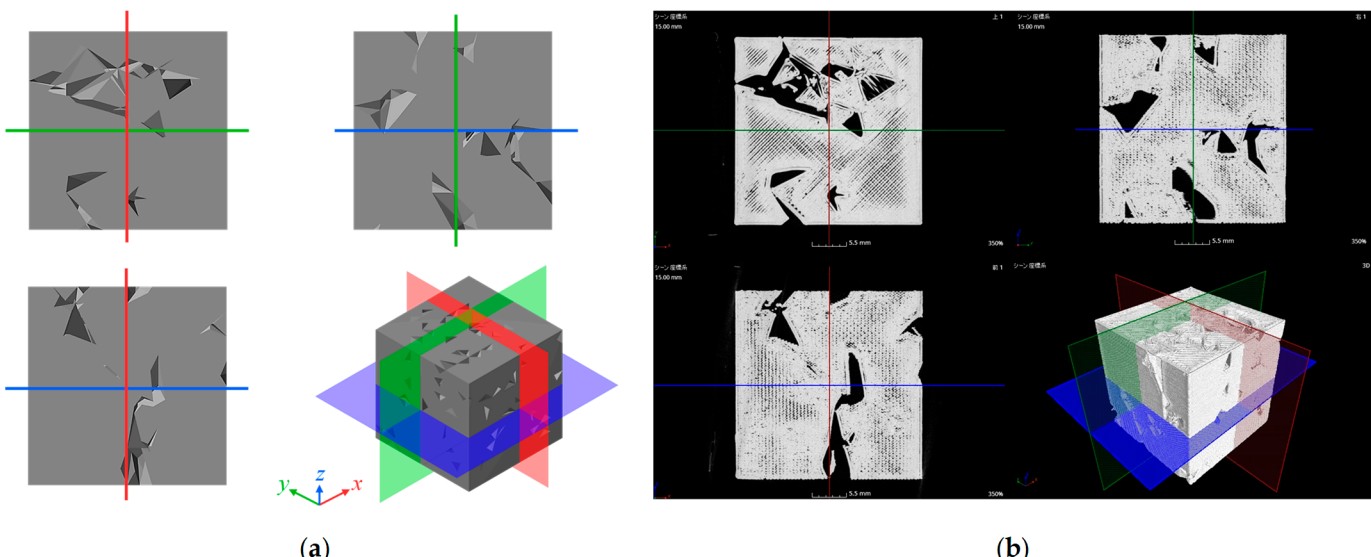

(**a**)  (**b**)

**Figure 20.** Images of the porous structure on different orthogonal planes. (**a**) Cross-sections of the CAD model of the specimen. (**b**) Cross-sections of the printed specimen obtained by micro-CT scan.

**Table 4.** Micro-CT-driven accuracy estimation at different layers of the porous structure.

| Height [mm] | 1 | 5 | 10 | 15 | 20 | 25 |
|---|---|---|---|---|---|---|
| *E* [%] | 2.86 | 3.55 | 4.06 | 3.55 | 2.18 | 3.12 |

In synopsis, images obtained from micro-CT scans provide more realistic results for complex structures. For micro-CT scans, the color of the printed image is not a problem; the same optimal image processing settings can be used for different colors of the printed objects.

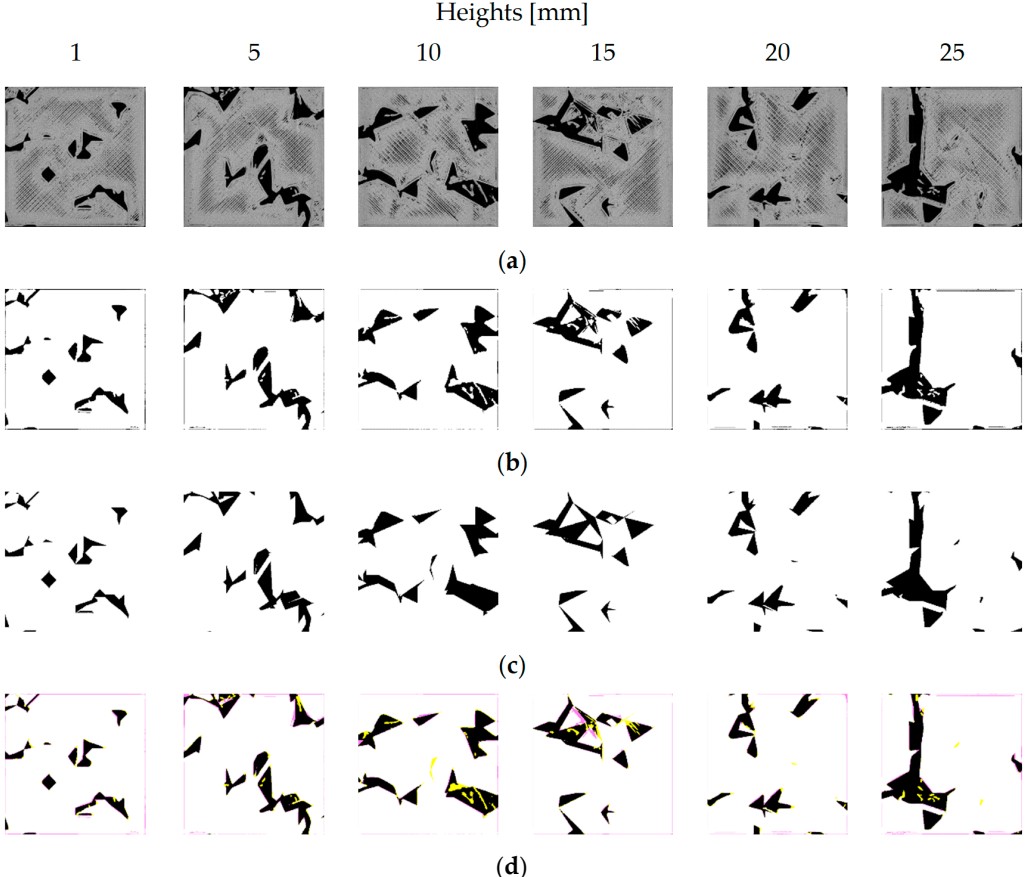

**Figure 21.** Accuracy assessment using micro-CT-scan-driven images of the specimen. (**a**) Micro-CT scan-driven images of the specimen at different heights. (**b**) Processed images of the specimen for comparison. (**c**) Design images. (**d**) Comparison images.

## 6. Concluding Remarks

This article presents a custom-made image processing system. The system can generate binary images of a given CAD model. In addition, it can generate images of the CAD model's 3D-printed counterparts. Moreover, it can compare these two types of images pixel by pixel to confirm whether or not the 3D-printed objects comply with the CAD model.

Using the system, a user can see how the accuracy varies due to the image processing settings such as the grayscale to binary image conversion threshold, noise reduction parameters, masking parameters, and pixel-fineness adjustment parameters.

It is found that the grayscale to binary image conversion threshold affects the accuracy most. In addition, the optimal threshold depends on the color of the 3D-printed object. Control over noise elimination during image processing (e.g., removing marks of the filaments on a given cross-section of a 3D-printed object) makes accuracy-checking more reliable.

The presented system can reliably measure the accuracy of not only 3D-printed objects with simple geometry but also 3D-printed objects with complex geometry (porous structures with random pore size, distribution, and depth). This was confirmed by the performance of case studies.

A simple object can exhibit an error of approximately 3%, even though visual inspection reveals that there is apparently no error in the 3D-printed object compared to its CAD model. A complex object can exhibit an error of approximately 10%, even though visual inspection reveals that the 3D-printed object has slight or no error compared to its CAD model; the error can decrease to approximately 2% when the presented system processes micro-CT scans and compares them with the images of the CAD model.

By leveraging the outcomes of this study, more pragmatic systems for the metrology of additive manufacturing can be developed.

**Author Contributions:** Conceptualization, S.U.; methodology, T.O. and S.U.; software, T.O. and S.U.; validation, T.O.; formal analysis, T.O. and S.U.; investigation, T.O. and S.U.; resources, S.U.; data curation, T.O. and S.U.; writing—original draft preparation, T.O. and S.U.; writing—review and editing, T.O. and S.U.; visualization, T.O.; supervision, S.U.; project administration, S.U.; funding acquisition, S.U. All authors have read and agreed to the published version of the manuscript.

**Funding:** This research received no external funding.

**Institutional Review Board Statement:** Not applicable.

**Data Availability Statement:** The data are available upon request to the corresponding authors.

**Acknowledgments:** The authors express their sincere gratitude to Shota Yonehara for his outstanding contributions to developing the system presented in this article.

**Conflicts of Interest:** The authors declare no conflicts of interest.

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
