# Peer review of "Verifying the Accuracy of 3D-Printed Objects Using an Image Processing System"

_jmmp, doi:10.3390/jmmp8030094_

Round 1

Reviewer 1 Report

Comments and Suggestions for Authors

This paper presented an image processing system for measuring the errors of 3D printed objects. The proposed image processing system compares the binary images of the printed object with the CAD model STL data pixel by pixel to generate an error metric for indicating the accuracy. The binary image of the 3D printed object is generated from the original image through a process of converting to grayscale image using user defined RGB values and applying user defined threshold for noise removal, mask operation, and binary conversion. A case study of comparing a simple 3D printed object with its 3D CAD model studied the image processing settings on the accuracy of the system. It shows that the grayscale to binary image conversion threshold is the most important parameter that affects the accuracy of the system. An optimal grayscale to binary image conversion threshold found with the simple 3D printed object is used in the case study of a complex 3D printed object with internal porosity. The system can reliably measure the accuracy of 3D printed objects with complex geometry, especially if the image is taken with micro-CT scanning. The work of this paper can be used to develop additive manufacturing metrology. A good review of the current technology has been conducted on evaluating the accuracy of 3D printed objects. The paper is well written and can be published as it is.

Author Response

We gratefully acknowledge the reviewer for the invaluable comments and suggestions. Our response is attached.

Reviewer 2 Report

Comments and Suggestions for Authors

The article focusses on the using a custom made image processing system for the use of verification of accuracy of 3D printing. Being prone to a certain amount of inaccuracy there is a need for good quality control techniques.

In general the article is nicely written and structured. 

Some comments:

2. related work, line 79 -120: the authors spend a great deal on explaining that 3D printing can lead to product defects, however it would have helped to have a clear overview of the possible quality control techniques instead of a text with examples of inaccuracies as well as techniques. Suggestion: skip most of the examples of inaccurate 3D and give structured info on the control techniques that can be used 

Although good description of image processing is provided I miss the involvement of machine learning (for which data augmentation/pre-processing are neccessary in image processing) and which could lead to improved comparison and assessment of defects

section 5:

line 407: process has been interrupted to take pictures: can authors guarantee that this has not negatively influenced quality of the printed item? (especially as they used an extrusion technique)

Author Response

(The authors gave the same response as above.)

Reviewer 3 Report

Comments and Suggestions for Authors

The authors presented an interesting work on in-situ verifying the accuracy of a 3D-printed object involves using an image processing system. However, there are few areas of improvement to be addressed.

1) Sect. 2 may need to reduce as the literature survey was lengthy with 2 pages. Try to highlight the key points and gaps for benefits of readers.

2) Fig. 15, instead of labelling 'Green' & 'Orange' only. The labels should include the definitions of both colours, even though they had defined in earlier section.

Comments on the Quality of English Language

Quality of English is technically well-sounded. Easy to read

Author Response

(The authors gave the same response as above.)
